# Effect of Zanthoxylum bungeanum essential oil on rumen enzyme activity, microbiome, and metabolites in lambs

**Hailong Zhang[1,2], Xia Lang[2,3], Xiao Li[1], Guoshun Chen[1]\*, Cailian Wang[2,3]\***

**1** College of Animal Science and Technology, Gansu Agriculture University, Lanzhou, China, **2** Key laboratory for Sheep, Goat and Cattle Germplasm and Straw Feed in Gansu Province, Lanzhou, China, **3** Institute of Animal Science and Grass Science and Green Agriculture, Gansu Academy of Agricultural Sciences, Lanzhou, China

\* chengs@gsau.edu.cn (GC); wangcl1974@163.com (CW)

## Abstract

Antibiotics were once used in animal production to improve productivity and resistance to pathogenic microbiota. However, due to its negative effects, the search for a new class of substances that can replace its efficacy has become one of the urgent problems to be solved. Plant essential oils (EOs) as a natural feed additive can maintain microbiota homeostasis and improve animal performance. However, its specific mechanism of action needs to be further investigated. Therefore, we added different doses of essential oil of *Zanthoxylum bungeanum* (EOZB) to the diets of Small Tail Han Sheep hybrid male lambs (STH lambs) to evaluate the effect of EOZB on rumen enzyme activity, rumen microbiology, and its metabolites in STH lambs. Twenty STH lambs were randomly divided into four groups (n = 5/group) and provided with the same diet. The dietary treatments were as follows: basal diet (BD) group; BD+EOZB 5 ml/kg group; BD+EOZB 10 ml/kg group; BD+EOZB 15 ml/kg group. We found that EOZB 10 ml/kg helped to increase rumen pectinase ($P<0.05$) and lipase ($P<0.05$) activities. Microbial 16S rRNA gene analysis showed that EOZB significantly altered the abundance of rumen microbiota ($P<0.05$). LC/GC-MS metabolomic analysis showed that the addition of EOZB produced a total of 1073 differential metabolites, with 58 differential metabolites remaining after raising the screening criteria. These differential metabolites were mainly enriched in glycerophospholipid metabolism, choline metabolism in cancer, retrograde endocannabinoid signaling, benzoxazinoid biosynthesis, and protein digestion and absorption. Correlation analysis showed that some rumen microbiota were significantly correlated with differential metabolite and enzyme activities.

## Introduction

Microbiota is vital to the health of the host [1]. The mammalian gastrointestinal tract is inhabited by $10^{13}$ to $10^{14}$ microbiota that plays an important role in host immune regulation, nutrient metabolism, maintenance of the structural integrity of the intestinal barrier, and defense

**Data Availability Statement:** A AVAILABILITY STATEMENT Metabolome raw sequence data were uploaded to the MetaboLights database and are available through accession number MTBLS3683. Microbiome raw sequence data were uploaded to

the National Center for Biotechnology Information (NCBI) database and are available through accession number SRR16774443- SRR16774622.

**Funding:** We thank for the financial support: No. 1760683; No.18YF1NA091; No. 18JR2RA032; The funder, Cailian Wang, played an important role in the study design design. The funder, Guoshun Chen, played an important role in the decision to publish.

**Competing interests:** The authors declare that the research was conducted in the absence of any commercial or financial relationships that could be construed as a potential conflict of interest.

against pathogens [2, 3]. The homeostasis of the gastrointestinal microbiota is relevant to host health [4, 5]. In animal production, plant EOs are used as a natural feed additive to maintain microbiota homeostasis and improve animal performance [6–8]. However, its specific mechanism of action needs to be further investigated.

Antibiotics were once used in animal production to improve animal productivity and maintain health status. However, due to their negative effects such as drug resistance and food safety concerns, other strategies have been attempted to improve animal growth performance and immune function [9]. For example, astragalus by-products had beneficial effects on rumen fermentation patterns and lipid metabolism in sheep, with no negative effects on their production performance and humoral immunity [10]. Direct feeding of probiotics to cows stabilizes rumen pH, the reduction in $CH_4$ production and the effect on the milk fatty acid profile of cows depending on strain type and diet formulation [11]. Feeding oregano oil to sheep did not change rumen pH, but promoted the growth of certain rumen microbiota, and low doses of oregano oil had a positive effect on rumen microbes, while high doses harmed rumen microbes [12]. In a dextran sodium sulfate-induced mouse model, essential oil of Zanthoxylum bungeanum pericarp alters the level of colonic microbial composition and has a positive effect on ulcerative colitis [13]. The above studies suggest that herbs, probiotics, and essential oils are some of the strategies currently used in animal production to promote growth and improve immunity. Although these studies have yielded fruitful results, there are some shortcomings. The health status of the animal and the increase in productivity are the result of the collaboration of multiple systems in the organism and are not determined by a few single systems. Therefore, in this study, the effects of EOZB on rumen enzyme activity, rumen microbial composition, and its metabolites in STH lambs were analyzed by a combination of kit method, 16s rRNA high-throughput sequencing, and LC/GC-MS metabolomics, aiming to provide new insights into the mechanism of action of EOZB in ruminants.

## Materials and methods

### Ethics statement

All studies involving animals were carried out under the regulations for the Administration of Affairs Concerning Experimental Animal (Ministry of Science and Technology, China; revise in June 2004), and sample collection protocols were approved by the Experimental Animal Ethics Committee of Gansu Agricultural University (Approval No. GSAU-Eth-AST-2021-026). Written informed consent was obtained from the owners for the participation of their animals in this study.

### Experimental design and sampling

Twenty STH lambs (male lambs, 3-month-old, initial weight = 23.57 ± 4.61 kg, $P>0.05$) were obtained from a farm in Dingxi City (Gansu Province, China). The experimental lambs were housed in individual pens that were disinfected and cleaned regularly. The lambs were randomly divided into four groups (n = 5/group) and provided the same diets (Table 1). The BD (quality = 1 kg) was provided for each lamb every day. ALW group was fed a BD; BLW group: BD+ EOZB 5 ml/kg; CLW group: BD+ EOZB 10 ml/kg; DLW group: BD+ EOZB 15 ml/kg. According to different trial groups, different doses of EOZB were administered daily. The experiment was performed for a total of 52 days, the lambs in each group were stunned (head electric shock) and then sacrificed (performed by professional butchers) by exsanguination of the jugular vein after the end of the trial. Rumen content (solid-liquid mixing) was collected into 1.5 ml sterile polypropylene tubes, immediately frozen in liquid nitrogen at -80 ˚C, and stored for further analysis of the microbiome and metabolome. At the same time, rumen

**Table 1. Basal diet composition and nutritional level (DM basis).**

| Dietary composition | % |
| --- | --- |
| Wheat straw | 50 |
| Concentrate pellets | 50 |
| Nutritional levels | |
| CP | 19.87 |
| EE | 5.06 |
| Ash | 16.85 |
| Ga | 1.32 |
| P | 0.60 |
| NDF | 113.50 |
| ADF | 61.89 |

Note: The concentrated pellet was purchased from the manufacturer (5, 10 and 15 ml/kg of EOZB were added), and nutrient levels were determined in the laboratory.

content was collected in sterilized cryotubes and stored at -20 ˚C for the determination of rumen digestive enzyme activity.

## Rumen digestive enzyme activity

The activity of eight rumen digestive enzymes (cellulase, neutral xylanase, pectinase, lysozyme, neutral protease, lipase, α-amylase, and pepsin) was determined. Each sample underwent a homogenization (0.9% normal saline) pretreatment, with 1 g of frozen rumen content being accurately weighed and added to the homogenization medium at a mass-volume ratio of 1:9 (g/mL). Mixing took place in an ice water bath to make a 10% homogenate. The homogenate was then centrifuged (4000 rpm for 10 min at room temperature), the supernatant was collected, and the target enzyme activity was determined according to the instructions of the appropriate kit. Total protein was measured by the total protein detection kit (Nanjing Jiancheng Bioengineering Institute, Jiangsu, China). The activities of cellulase, neutral xylanase, pectinase, lysozyme, neutral protease, lipase, α-amylase, and pepsin were measured by the cellulase detection kit (Nanjing Jiancheng Bioengineering Institute, Jiangsu, China), the neutral xylanase detection kit (Beijing Solarbio Science & Technology Co., Ltd, Beijing, China), the pectinase detection kit (Nanjing Jiancheng Bioengineering Institute, Jiangsu, China), the lysozyme detection kit (Nanjing Jiancheng Bioengineering Institute, Jiangsu, China), the neutral protease detection kit (Beijing Solarbio Science & Technology Co., Ltd, Beijing, China), the lipase, α-amylase detection kit (Nanjing Jiancheng Bioengineering Institute, Jiangsu, China), and the pepsin detection kit (Nanjing Jiancheng Bioengineering Institute, Jiangsu, China).

## DNA extraction and high-throughput sequencing

Total DNA was extracted from ruminal samples using the E.Z.N.A.® Stool DNA Kit (Omega Bio-Tek, Norcross, GA, USA) according to the manufacturer's protocol. The V4-V5 region of the bacteria 16S ribosomal RNA gene was amplified by PCR (95 ˚C for 5 min, followed by 30 cycles at 95 ˚C for the 30 s, 55 ˚C for 30 s, and 72 ˚C for 45 s, and a final extension at 72 ˚C for 5 min), using primers 515F (5ʹ-GTGCCAGCMGCCGCGG-3ʹ) and 907R (5ʹ-CCGTCAATTCMTTTRAGTTT-3ʹ), where the barcode was a six-base sequence unique to each sample. PCR reactions were performed in 30 μl of the mixture containing 15 μl of 2 × Phanta Master Mix, 1 μl of each primer (10 μM), and 20 ng of template DNA. Amplicons were

extracted from 2% agarose gels and purified using the AxyPrep DNA Gel Extraction Kit (Axygen Biosciences, Union City, CA, USA), according to the manufacturer's instructions. Purified PCR products were quantified by Qubit®3.0 (Life Invitrogen, CA, USA) and every 20 amplicons whose barcodes were different were mixed equally. The pooled DNA product was used to construct the Illumina pair-end library following Illumina's genomic DNA library preparation procedure. The amplicon library was then paired-end sequenced (2×250) on an Illumina Novaseq 6000 platform (Nanjing GenePioneer Co. Ltd, Nanjing, China) according to the standard protocol.

Raw data returned by the Illumina HiSeq sequencing platform were filtered using the software packages Pandaseq [14], PRINSEQ [15], and Vsearch [16] (v2.15.0_linux_x86_64) to remove chimeras and obtain optimized sequences (tags). Operational taxonomic units (OTUs) were clustered using Vsearch (v.2.15.0_linux_x86_64), and the clustering similarity threshold was 97%. Using the self-developed perl program to random rarefaction the data of each sample (the number of rarefaction is the minimum number of sample sequences). QIIME (v.1.9.1) was used to select the most abundant sequence from each OTU as the representative sequence, and then the Uclust method was used to compare the representative sequence to the Silva rRNA database (release_132) and classify the OTU species. Based on the abundance and annotation information of OTUs, the proportion of sequences in each sample at different taxonomic levels was counted to assess species abundance and species diversity of the samples. Alpha diversity was assessed by Observed species, Chao1, Shannon, and Simpson indexes [17], and data were plotted as a rarefaction curve. Beta diversity analysis was used to compare differences in species diversity (microbial composition and structure) between different samples. Using PICRUSTt2 [18] for functional gene analysis (The OTU abundance table was standardized, and the KO information and COG family information corresponding to OTUs were obtained by mapping each OTU's corresponding relative, aligning to the KEGG and COG databases), which predicts the gene function of the sample and calculate the functional gene abundance.

## LC-MS analysis

Metabolomics analysis was performed on a dual detection platform. First, perform metabolomics analysis based on an LC-MS system combined quadrupole Orbitrap mass spectrometer (Q Exactive Orbitrap, Thermo Fisher Scientific, USA.) [19]. STH lamb rumen contents that had been preserved at -80˚C (20 cases; 4 groups with 5 cases in each group) were used to perform LC-MS metabolomic analysis. A 50 mg sample was weighed and mixed with 1000 μl of extraction solution (methanol: acetonitrile: water = 2:2:1 (V/V), containing isotope-labeled internal standard mixture); grinding took place at 35 Hz for 4 min, followed by sonication in an ice water bath for 5 min (repeated 3 times), then to -40˚C let stand for 1 h. The samples were centrifuged at 4 ˚C, 12 000 RPM [centrifugal force 13800 (×g), radius 8.6 cm] for 15 min. The supernatant was taken up in an injection vial for machine detection, all samples with an equal amount of supernatant were mixed into a QC sample for machine detection.

A Vanquish ultra-high performance liquid chromatograph (Thermo Fisher Scientific) was used to separate the target compounds through a Waters ACQUITY UPLC BEH Amide (2.1 mm × 100 mm, 1.7 μm) liquid chromatographic column. The A phase of the liquid chromatography was the water phase, containing 25 mmol/l ammonium acetate and 25 mmol/l ammonia water, and the B phase was acetonitrile. A sample pan temperature of 4˚Cand an injection volume of 3 ml was used. A Thermo Q Exactive HFX mass spectrometer performed primary and secondary mass spectrometric data acquisition under the control of Xcalibur software (v.4.0.27, Thermo). The detailed parameters were as follows: sheath gas flow rate: 30

ARB, aux gas flow rate: 25 ARB, capillary temperature: 350 ˚C, full MS resolution: 60 000, MS/MS resolution: 7500, collaboration energy: 10/30/60 in NCE mode, spray Voltage: 3.6 kV (positive) or—3.2 kV (negative).

After the original data was converted into mzXML format by ProteoWizard software, a self-written R program package (the kernel was XCMS) was used for peak identification, peak extraction, peak alignment, and integration, and then it was matched with the second mass spectrum database for substance annotation and algorithm scoring. The Cutoff value was set to 0.3. After data quality control and data preprocessing of metabolomics raw data, differential metabolite screening and identification, KEGG annotation of differential metabolites, KEGG pathway type analysis of differential metabolites, and KEGG pathway enrichment analysis of differential metabolites were carried out.

## GC-TOF-MS analysis

Metabolomics analysis was performed on a dual detection platform. Secondly, perform metabolomics analysis based on a GC-MS gas chromatography-time-of-flight mass spectrometer equipped with an Agilent DB-5MS capillary column (J&W Scientific, Folsom, CA, USA). STH lamb rumen content, preserved at -80˚C (20 cases;4 groups with 5 cases in each group) was used to perform GC-TOF-MS metabolomic analysis. Samples of 50 ± 1 mg were taken in a 2 ml EP tube and 1000 μl pre-cooled extraction solution (methanol acetonitrile-water volume ratio = 2:2:1) containing 5 μl ribitol. The mixture was vortexed for 30 s, steel beads were added, 35 Hz grinding took place for 4 min, followed by sonication in an ice water bath for 5 min (repeated 3 times), then to -40˚C let stand for 1 h. The samples were centrifuged at 4 ˚C, 12 000 RPM [centrifugal force 13800 (×g), radius 8.6 cm] for 15 min and 400 μl of supernatant was carefully pipetted into a 1.5 ml EP tube. Samples of 50 μl were taken and mixed into a QC sample. The extract was dried in a vacuum concentrator and 30 μl of methoxyamine salt reagent (methoxyamine hydrochloride, dissolved in pyridine 20 mg/ml) was added to the dried metabolites. After mixing gently, the mixture was placed in an oven and incubated at 80˚C for 30 min. Aliquots of 40 μl BSTFA (containing 1% TMCS, v/v) were added to each sample, the mixture was incubated at 70˚C for 1.5 h, cooled down to room temperature, 5 μl of FAMEs (dissolved in chloroform) were added to the mixed sample, and random order testing was conducted.

GC-TOF-MS analysis was performed using an Agilent 7890 gas chromatograph coupled with a time-of-flight mass spectrometer. The system utilized a DB-5MS capillary column. Aliquots (1 μl) of the sample were injected in splitless mode. Helium was used as the carrier gas, the front inlet purge flow was 3 ml/min, and the gas flow rate through the column was 1 ml/min. The initial temperature was kept at 50˚C for 1 min, then raised to 310˚C at a rate of 10˚C/min, then kept for 8 min at 310˚C. The injection, transfer line, and ion source temperatures were 280˚C, 280˚C, and 250˚C, respectively. The energy was -70 eV in electron impact mode. The mass spectrometry data were acquired in full-scan mode with an m/z range of 50–500 at a rate of 12.5 spectra per second after a solvent delay of 6.27 min.

Mass spectrometric data were analyzed using ChromaTOF software (v.4.3x, LECO) for peak extraction, baseline correction, deconvolution, peak integration, and peak alignment [20]. In the qualitative work of substances, the LECO-Fiehn Rtx5 database was used, including mass spectrometry matching and retention time index matching. Finally, peaks with a call rate below 50% or RSD >30% in QC samples were removed [21]. After data quality control and data preprocessing of metabolomics raw data, differential metabolite screening and identification, KEGG annotation of differential metabolites, KEGG pathway type analysis of differential

metabolites, and KEGG pathway enrichment analysis of differential metabolites were carried out.

## Statistical data analysis

SPSS software (v.21.0, SPSS Inc.) was used for one-way analysis of variance to analyze the relative abundance of rumen bacteria and the activity of digestive enzymes in the rumen at different levels, multiple-comparisons between groups were performed using Duncan's method, $P < 0.05$ indicated a significant difference. R (v.3.6.2) was used to perform rank-sum test analysis on different indexes of rumen microbial Alpha diversity; $P < 0.05$ indicated a significant difference. Spearman correlation test was used to analyze the correlation between rumen microbes with rumen metabolites and rumen enzyme activity. Metabolomics data conduct univariate statistical analysis (including Student's t-test) and multivariate statistical analysis (including principal component analysis (PCA) and orthogonal partial least squares (OPLS-DA) analysis).

## Results

### Rumen enzyme activities

The effect of EOZB on ruminal enzyme activity of STH lambs is shown in Table 2. The activity of the digestive enzymes measured tended to increase and then decrease with increasing doses of EOZB. Among them, EOZB 10 ml/kg (CLW group) significantly increased the activity of rumen Pectase and Lipase ($P < 0.05$).

### Diversity of rumen microbes

A total of 2727562 tags were obtained from 20 samples in this study, and 2552906 clean tags were obtained after stitching, optimization, and quality control. According to the minimum number of detected sequences, 117434 sequences were taken from each sample for subsequent analysis (S1 Table). Clustering was performed at 0.97 similarities using Vsearch, resulting in OTU for each sample. A total of 3677 OTUs were generated, and 2940 remained after random rarefaction, and representative sequences of OTUs were selected for species annotation. There were 2600 shared OTUs in ALW, BLW, CLW, and DLW, 2829 OTUs in ALW and BLW, 2914 OTUs in ALW and CLW, and 2908 OTUs in ALW and DLW. The number of unique OTUs in the CLW group was significantly higher than that in the other groups (Fig 1A). Species accumulation curves, which are used to describe the condition of increasing species as sample size

**Table 2. Effect of EOZB on enzyme activity in the rumen content of STH lambs (U/mg protein).**

| Digestive enzyme | ALW | BLW | CLW | DLW | P-Value |
|---|---|---|---|---|---|
| Cellulase | 1351.01±38.82 | 1293.82±59.35 | 1402.91±34.68 | 1029.36±22.97 | 0.306 |
| Neutral xylanase | 688647.87±928.28 | 806880.53±635.70 | 1058229.64±566.41 | 563015.84±927.08 | 0.099 |
| Pectase | 1.36±0.02c | 1.74±0.01b | 1.98±0.03a | 1.35±0.02c | <0.001 |
| Lysozyme | 798.71±66.63 | 857.42±75.61 | 970.15±81.95 | 645.66±93.36 | 0.109 |
| neutral protease | 48287.97±212.43 | 55163.29±590.40 | 63302.75±275.63 | 46436.71±198.58 | 0.329 |
| Lipase | 466005.35±33.54c | 552713.45±45.77b | 555095.54±51.29a | 377892.87±41.54d | <0.001 |
| Amylase | 0.87±0.05 | 0.98±0.09 | 1.08±0.03 | 0.73±0.07 | 0.052 |
| Pepsin | 103.04±3.14 | 118.35±1.51 | 119.11±6.81 | 90.02±5.70 | 0.195 |

Note: Different lowercase letters in a row indicate significant differences ($P < 0.05$).

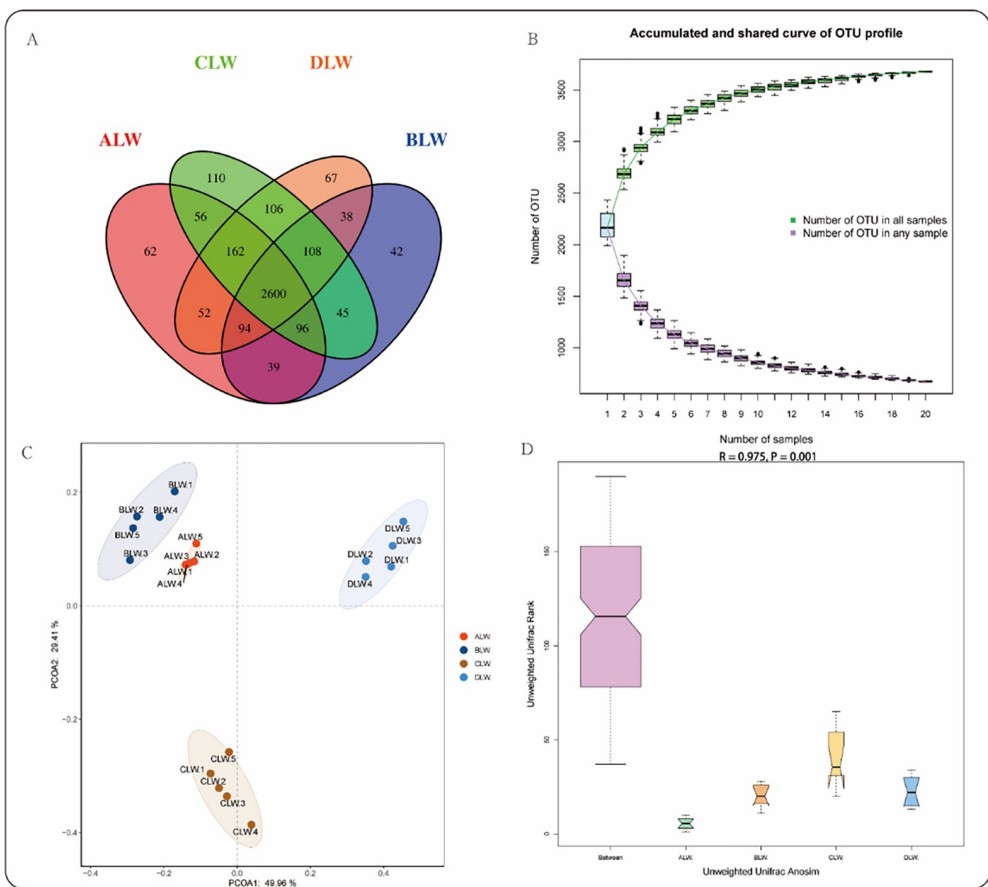

**Fig 1. Differences in the structure of STH rumen bacteria with the addition of EOZB.** (A) Analysis of OTU-Venn diagram of each experimental group with EOZB; (B) Species accumulation curves; (C, D) PCoA and Anosim analysis.

increases, are widely used to judge the adequacy of sample size and for the estimation of species richness. The curve began to flatten as the sample size reached 10, indicating adequate sample size and saturation of sequencing coverage (Fig 1B). PCoA analysis revealed significant differences in rumen microbial species between the EOZB group and the BD group (Fig 1C). Anosim analysis further showed that the differences between groups were significantly greater than within groups (Fig 1D). As shown in Table 3, the Chao1 index of EOZB 15 ml/kg (DLW group) was significantly higher ($P<0.05$) than the other trial groups, indicating that a certain dose of EOZB could alter the rumen microbial abundance of STH lambs. Shannon and Simpson indices were also higher in the EOZB 15 ml/kg (DLW group) than in the control group, but the difference was not significant.

## Effect of EOZB on rumen microbial composition

At the taxonomic level, the phylum level Bacteroidetes, Firmicutes, Proteobacteria, Tenericutes, Fibrobacteres, and Spirochaetes were the dominant bacteria, with their relative abundance greater than 1%. The relative abundance of Bacteroidetes and Firmicutes had the highest relative abundance in each trial group, accounting for more than 88% of rumen bacteria (Fig 2A). In addition, among the bacteria with relative abundance greater than 1% at the

**Table 3. Effect of EOZB on the index of ruminal alpha diversity in STH lambs.**

| Index | ALW | BLW | CLW | DLW | SEM | P-Value |
|---|---|---|---|---|---|---|
| Chao1 | 2415.65b | 2413.15b | 2411.76b | 2521.18a | 10.17 | <0.001 |
| Observed species | 2155.60 | 2151.20 | 2131.00 | 2151.60 | 19.05 | 0.579 |
| Shannon | 8.24 | 8.26 | 8.25 | 8.27 | 0.04 | 0.851 |
| Simpson | 0.99 | 0.99 | 0.99 | 0.99 | <0.01 | 0.582 |

Note: Different lowercase letters in a row indicate significant differences ($P<0.05$).

rumen phylum level, Tenericutes, Fibrobacteres, and Spirochaetes had the highest relative abundance in the ALW group ($P<0.05$); Firmicutes had the highest relative abundance in the BLW group ($P<0.05$); Bacteroidetes had the highest relative abundance in the CLW group ($P<0.05$); the highest relative abundance of Proteobacteria in the DLW group ($P<0.05$) (S2 Table). Genus level *Prevotella_1*, *Rikenellaceae_RC9_gut_group*, *Prevotellaceae_UCG_001*, *Prevotellaceae_UCG_003*, *Christensenellaceae_R_7_group*, *Fibrobacter*, *Rumen_bacterium_YS3*, *Ruminococcaceae_NK4A214_group*, *Succiniclasticum*, *Treponema_2*, *Saccharofermentans*, and *Ruminococcaceae_UCG_014* were the dominant genera and their relative abundance was greater than 1% (Fig 2B).

*Prevotella_1* and *Rikenellaceae_RC9_gut_groups* had the highest relative abundance in each trial group, accounting for more than 28% of the total rumen bacteria. In addition, among the bacteria with relative abundance greater than 1% at the rumen genus level, the ALW group *Prevotella_1*, *Rumen_bacterium_YS3*, *Succiniclasticum* had the highest relative abundance ($P<0.05$); the BLW group *Prevotellaceae_UCG_003*, the *Christensenellaceae_R_7_group*, *Saccharofermentans* showed the highest relative abundance, but only *Christensenellaceae_R_7_group* showed a significant level of relative abundance ($P<0.05$); CLW group *Rikenellaceae_RC9_gut_group*, *Ruminococcaceae_NK4A214_group*, *Treponema_2*, had the highest relative abundance ($P<0.05$); DLW group *Prevotellaceae_UCG_001*, *Fibrobacter* had the highest relative abundance, but the difference was not significant (S2 Table). LEfSe analysis of samples between groups showed that the addition of EOZB created a total of 21 differential biomarkers (LDA score >4) (Fig 3), which is consistent with ANOVA analysis. Overall, the addition of EOZB caused significant differences.

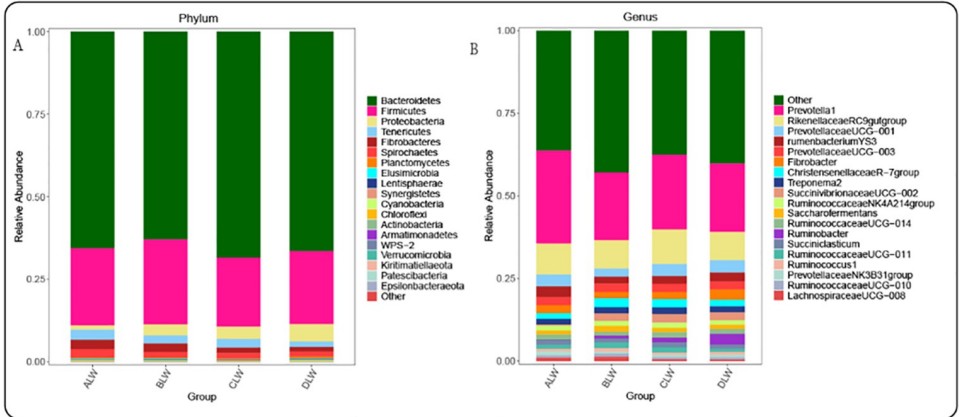

**Fig 2. Abundance of STH rumen bacteria added with EOZB at phylum and genus levels.** (A) Relative abundance of rumen bacteria at the phylum level; (B) Relative abundance of rumen bacteria at the genus level.

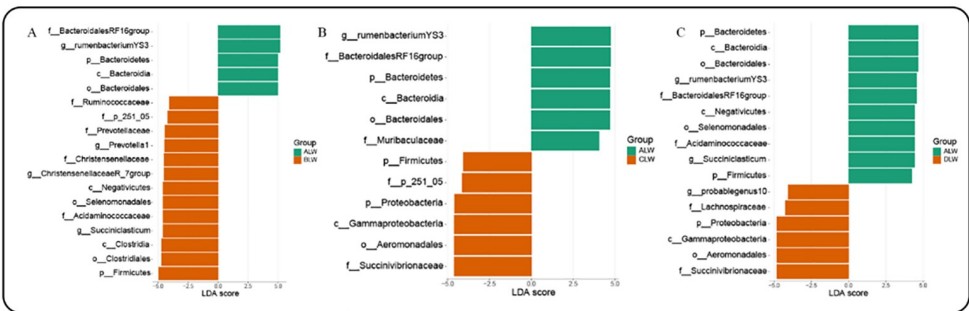

**Fig 3. Between-group biomarkers.** LDA value distribution histogram.

## Gene function prediction

Using PICRUSt2 to predict gene function, 38 KEGG gene families and 24 COG gene families were identified in 16S rRNA gene sequencing data. Among them, five KEGG gene families showed significant differences in the EOZB groups (Fig 4), and three COG gene families showed significant differences (Fig 5). The addition of low-dose (BLW) EOZB in the KEGG gene family significantly enhanced the Circulation system ($P$ = 0.009) and Infectious diseases:

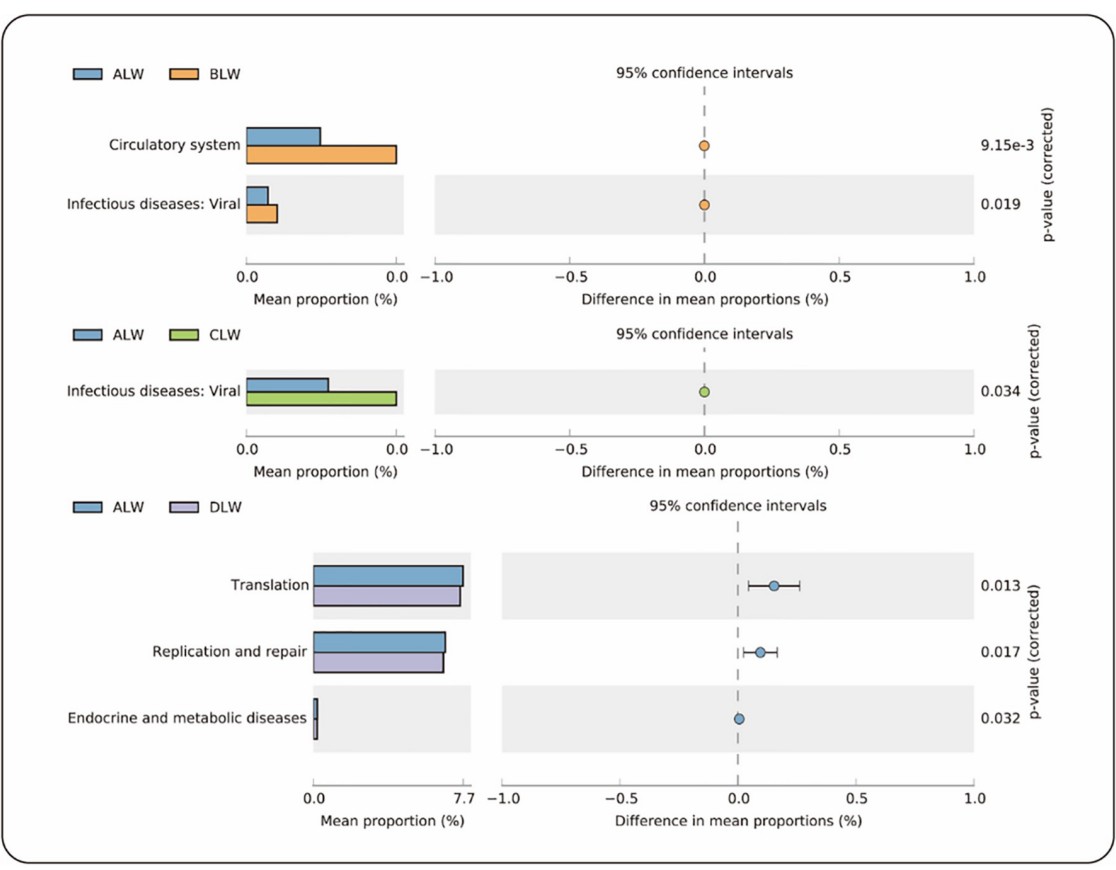

**Fig 4. KEGG functional pathways.** Differences in KEGG functional pathways between groups.

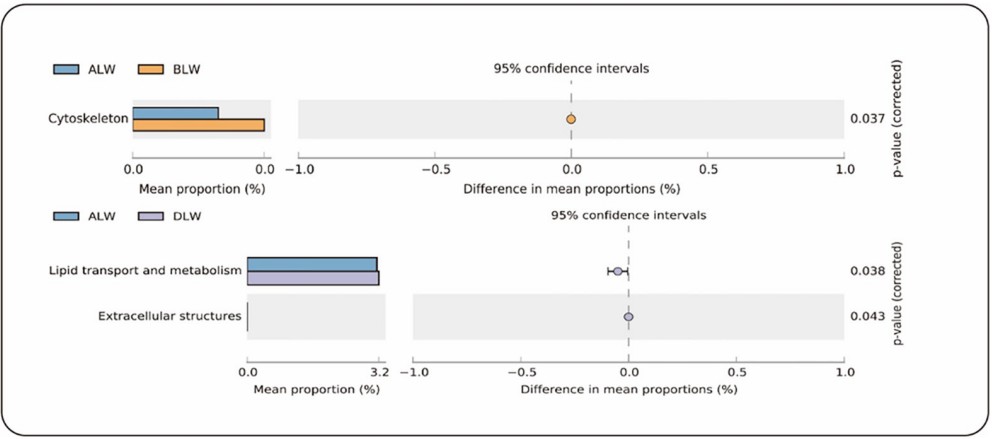

**Fig 5. COG functional pathways.** Differences in COG functional pathways between groups.

viral ($P$ = 0.019) related genes. The addition of medium-dose (CLW) EOZB significantly enhanced Infectious diseases: viral ($P$ = 0.034) associated genes. The addition of high-dose (DLW) EOZB significantly decreased Translation ($P$ = 0.013), Replication and repair ($P$ = 0.017), and Endocrine metabolic diseases ($P$ = 0.032) related genes. In the COG gene family, adding low-dose (BLW) EOZB significantly enhanced Cytoskeleton (P = 0.037) related genes. The addition of high-dose (DLW) EOZB significantly enhanced Lipid transport and metabolism ($P$ = 0.038) and Extracellular structures ($P$ = 0.043) related genes.

## Rumen metabolites and metabolic pathways

To further explore the effect of adding EOZB on rumen microbes and their metabolites in the studied STH lambs, the concentration of rumen metabolites in the four groups was analyzed by a combined method of LC-MS and GC-MS. A total of 1073 metabolites were found (S3 Table). These metabolites included amino acids, organic acids, lipids, nucleotides, etc. which are involved in many rumen biochemical processes in sheep. The PCA score chart was derived from the LC/GS-MS metabolic spectrum. From the results of the PCA score chart, all samples were within the 95% confidence interval; this data is worthy of further study (Fig 6A–6C). From the scatter plot of the OPLS-DA model, the sample distinction between groups was very significant (Fig 6D–6F). In addition, the OPLS-DA's permutation test showed that the original model did not have an over-fitting phenomenon, and the model was robust (the over-fitted models R2 and Q2 remained unchanged). Among them, the ALW group compared with BLW R2Y = 0.921, Q2Y = -0.345; ALW group compared with CLW R2Y = 0.936, Q2Y = 0.152; ALW group compared with DLW R2Y = 0.959, Q2Y = -0.012 (Fig 6G–6I). The above data demonstrated that the OPLS-DA model of this study was effective.

The differential metabolites between groups were screened according to the criteria of VIP >1.5 (VIP: the VIP value from OPLS-DA model), $P$ < 0.05. Among them, the ALW group had 28 different metabolites compared with BLW, including pe(20:0/14:1(9z)), prolylglycine, l-hypoglycin a, kynurenic acid, batatasin iii, n-lactoyl ethanolamine, beta-alanine, nicotine imine, 4-amino-2-methylenebutanoic acid, aspartyl-valine, lysope(0:0/18:3(6z,9z,12z)), asymmetric dimethylarginine, benzaldehyde, levan, pe(p-18:1 (11z)/14:1(9z)), 2,5-dihydro-2,4-dimethyloxazole, Indole, 2-pyridylacetic acid, l-allothreonine, pc(18:3(6z,9z,12z)/15: 0), 2-piperidinone, histidinyl-proline, pc(22:6(4z,7z,10z,13z,16z,19z)/16:0),

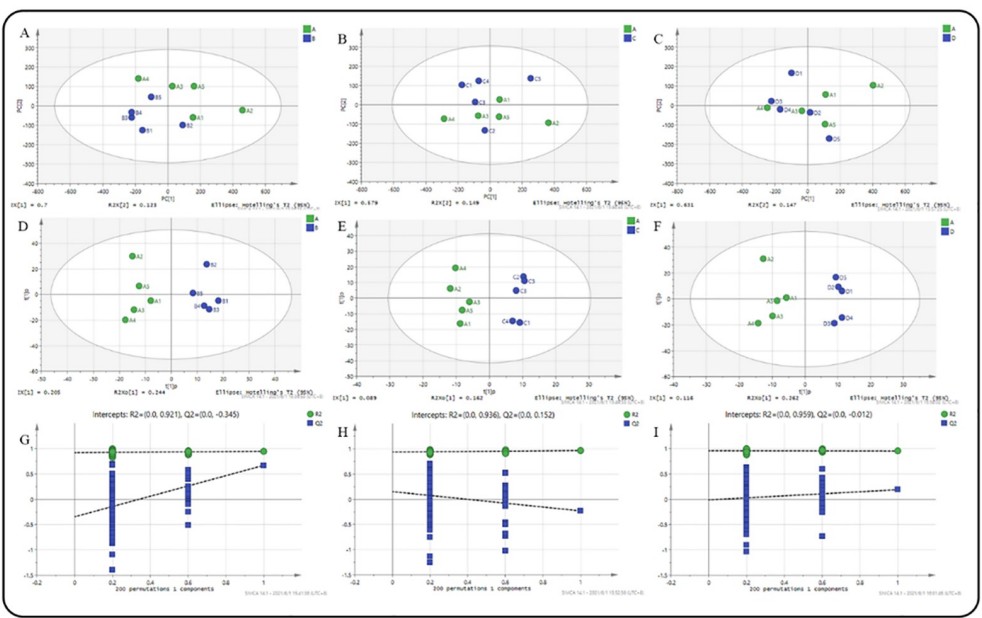

**Fig 6.** (A) Scatter plot of the PCA of ALW_vs_BLW; (B) Scatter plot of the PCA of ALW_vs_CLW; (C) Scatter plot of the PCA of ALW_vs_DLW; (D) Scatter point graph of the OPLS-DA model of ALW_vs_BLW; (E) Scatter point graph of the OPLS-DA model of ALW_vs_CLW; (F) Scatter point graph of the OPLS-DA model of ALW_vs_DLW; (G) Permutation test result of the OPLS-DA model of ALW_vs_BLW; (H) Permutation test result of the OPLS-DA model of ALW_vs_CLW; (I) Permutation test result of the OPLS-DA model of ALW_vs_DLW.

artomunoxanthentrione epoxide, n-acetyl-d-galactosamine 1, lysopc(20: 3(5z,8z,11z)), trans-3,3',4',5,5',7-hexahydroxyflavanone, and valyl-lysine. Compared with CLW, the ALW group had seven different metabolites, including methylsuccinic acid, pe(22:5(4z,7z,10z,13z,16z)/14:0), diethanolamine, pe(p-18:1(9z)/18:1(9z)), glutaconic acid, pc(18:3(6z,9z,12z)/15:0), and 2-piperidinone. Compared with DLW, the ALW group had 23 different metabolites, including pe(20:0/14:1(9z)), as 1–1, pc(18:1(11z)/p-16:0), hexacosanoyl carnitine, beta-alanine, pe(p-18:1(11z)/16:0), metenamine, asymmetric dimethylarginine, picraquassioside a, pe(p-18:1(11z)/14:1(9z)), pe(p-18:1(9z)/18:1(9z)), 2-propylfuran, n-cyclohexylformamide 1, pc(18:3(6z,9z,12z)/18:0), indole, muricatacin, digitoxose 1, pc(15:0/p-18:1(9z)), pc(18:3(6z,9z,12z)/15:0), 2-piperidinone, caprryloylglycine, artomunoxanthentrione epoxide, trans-3, and 3',4',5,5',7-hexahydroxyflavanone (S4 Table). Further enrichment analysis showed that adding EOZB to the diet significantly changed Glycerophospholipid metabolism, Choline metabolism in cancer, Retrograde endocannabinoid signaling, Benzoxazinoid biosynthesis, and Protein digestion and absorption (Fig 7A–7C).

## Correlation analysis

Spearman correlation coefficient analysis (r>0.5 or r<- 0.5, $P<0.05$) was performed using VIP>1.5 (VIP: the VIP value from OPLS-DA model) $P<0.05$ for differential metabolites with significantly different populations in abundance at the genus level ($P<0.05$). As shown in the Fig 8, *Prevotella_1* was significantly and positively correlated with pe(20:0/14:1(9z)), 2-piperidinone, indole, 4-amino-2-methylenebutanoic acid, pc(18:3(6z,9z,12z)/15:0), muricatacin, beta-alanine, pc(16:1(9z)/p-18:0), trans-3,3',4',5,5',7-hexahydroxyflavanone, aspartyl-valine, lysope(0:0/20:4(5z,8z,11z,14z)), artomunoxanthentrione epoxide, lysope(0:0/18:3(6z,9z,12z)),

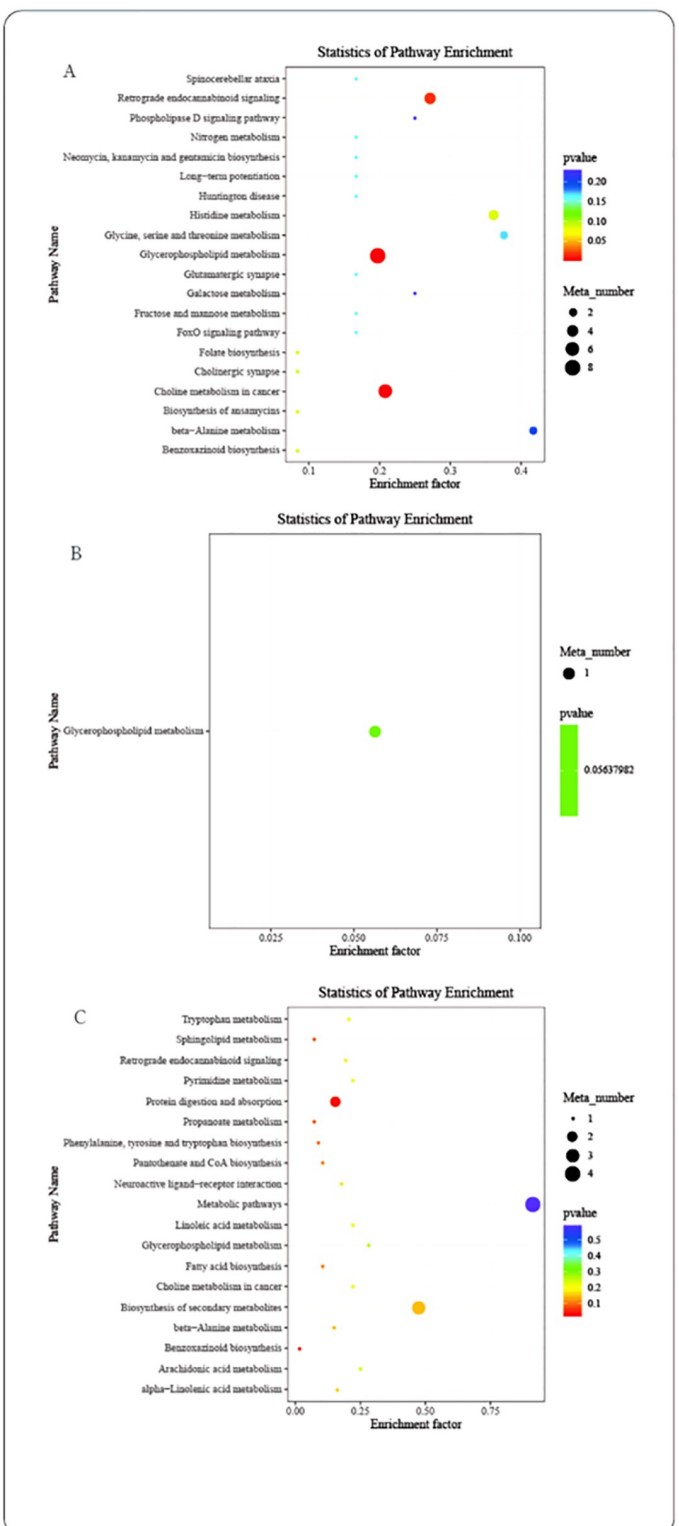

**Fig 7. Metabolic pathways.** Enrichment analysis of metabolic pathways.

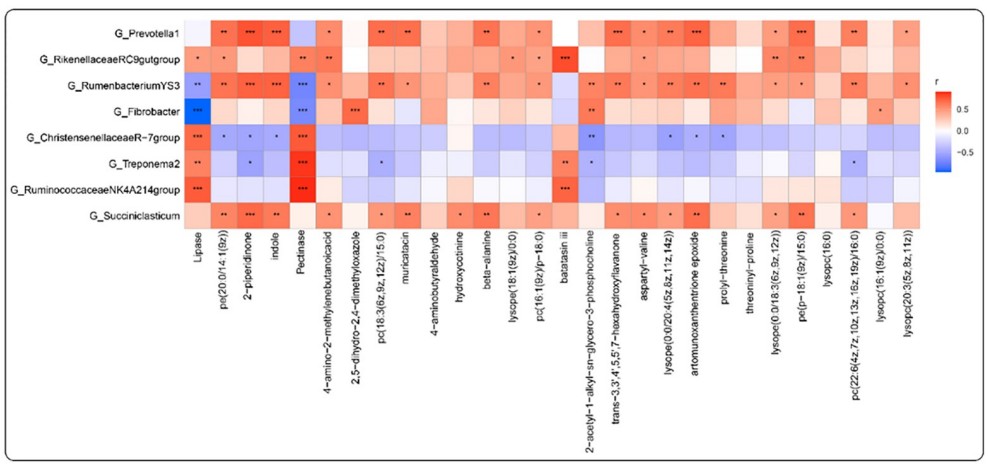

**Fig 8. The connection between microorganisms and metabolites and enzymatic.** Heatmap of correlations between microorganisms and metabolites and enzymatic activities. $^*P < 0.05$, $^{**}P < 0.01$, $^{***}P < 0.001$.

pe(p-18:1(9z)/15:0), pc(22:6(4z,7z,10z,13z,16z,19z)/16:0), lysopc(20:3(5z,8z,11z)) (r>0.5, P<0.05). *Rikenellaceae_RC9_gut_group* was significantly and positively correlated with pe (20:0/14:1(9z)), 4-amino-2-methylenebutanoic acid, lysope(18:1(9z)/0:0), pc(16:1(9z)/p-18:0), batatasin iii, aspartyl-valine, lysope(0:0/18:3(6z,9z,12z)), pe(p-18:1(9z)/15:0), pectinase, lipase (r>0.5,P<0.05). *Rumen_bacterium_YS3* was significantly and positively correlated with pe (20:0/14:1(9z)), 2-piperidinone, indole, 4-amino-2-methylenebutanoic acid, pc(18:3 (6z,9z,12z)/15:0), muricatacin, beta-alanine, pc(16:1(9z)/p-18:0), 2-acetyl-1-alkyl-sn-glycero-3-phosphocholine, trans-3,3',4',5,5',7-hexahydroxyflavanone, aspartyl-valine, lysope(0:0/20:4 (5z,8z,11z,14z)), artomunoxanthentrione epoxide, prolyl-threonine, lysope(0:0/18:3 (6z,9z,12z)), pe(p-18:1(9z)/15:0), pc(22:6(4z,7z,10z,13z,16z,19z)/16:0), lysopc(20:3(5z,8z,11z)) (r>0.5,P<0.05) and significantly negatively correlated with pectinase, lipase (r<0.5,P<0.05). *Fibrobacter* was significantly and positively correlated with 2,5-dihydro-2,4-dimethyloxazole, 2-acetyl-1-alkyl-sn-glycero-3-phosphocholine, lysope(18:1(9z)/0:0) (r>0.5,P<0.05) and significantly negatively correlated with pectinase, lipase (r<0.5,P<0.05). *Christensenellaceae_R_7_group* was significantly and negatively correlated with pe(20:0/14:1(9z)), 2-piperidinone, indole, 2-acetyl-1-alkyl-sn-glycero-3-phosphocholine, lysope(0:0/20:4 (5z,8z,11z,14z)), artomunoxanthentrione epoxide, prolyl-threonine (r<0.5,P<0.05) and positively correlated with pectinase, lipase (r>0.5,P<0.05). *Treponema_2* was significantly and negatively correlated with 2-piperidinone, pc(18:3(6z,9z,12z)/15:0), 2-acetyl-1-alkyl-sn-glycero-3-phosphocholine, pc(22:6(4z,7z,10z,13z,16z,19z)/16:0) (r<0.5,P<0.05) and significantly positively correlated with batatasin iii, pectinase, lipase (r>0.5,P<0.05). *Ruminococcaceae_NK4A214_group* was significantly and positively correlated with pectinase, lipase batatasin iii (r>0.5,P<0.05). *Succiniclasticum* was significantly and positively correlated with pe(20:0/14:1 (9z)), 2-piperidinone, indole, 4-amino-2-methylenebutanoic acid, pc(18:3(6z,9z,12z)/15:0), muricatacin, hydroxycotinine, beta-alanine, pc(16:1(9z)/p-18:0), trans-3,3',4',5,5',7-hexahydroxyflavanone, aspartyl-valine, lysope(0:0/20:4(5z,8z,11z,14z)), artomunoxanthentrione epoxide, lysope(0:0/18:3(6z,9z,12z)), pe(p-18:1(9z)/15:0), pc(22:6(4z,7z,10z,13z,16z,19z)/16:0) (r>0.5,P<0.05) (Fig 8).

## Discussion

*Z. bungeanum* is a plant belonging to the genus Zanthoxylum of the Rutaceae family. It is currently widely distributed in most parts of China and some Southeast Asian countries. *Z. bungeanum* is rich in flavonoids, including rutin, quercetin, foeniculin, hyperin, and isoquercitrin [22]. Owing to its unique flavor and medicinal value, *Z. bungeanum* is now widely used as a flavoring agent and traditional Chinese herbal medicine [23] as it has a variety of biological and pharmacological activities, including anti-inflammatory [24], antioxidant [25], and antibacterial properties [26]. EOZB is a thick paste-like fluid with a natural spicy flavor extracted from the husk of the plant and has a similar medicinal value to the whole plant. However, there are limited reports of its application in ruminants. In this study, STH lambs were used as experimental animals, and the effects of EOZB were studied from the perspective of rumen digestive enzyme activity, the microbiome, and metabolomics.

A large number of bacteria, fungi, archaea, and protozoa inhabit the rumen, which together constitutes a complex rumen micro-ecosystem [27]. These microbiota produce various digestive enzymes and play an important role in digestion. The activity of digestive enzymes mainly depends on the type and number of rumen microbial populations and is closely related to their metabolism [24]. Pectin is widely present in plant cell walls where it is used for adhesion and support [28]. Pectin can be degraded by pectinase, and the small molecules produced can be used by ruminants [29]. Many bacteria can produce lipase, which can hydrolyze fatty acids, especially long-chain fatty acids [30]. Our research results show that adding a medium dose of EOZB (10 ml/kg) significantly increased the activities of pectinase ($P<0.05$) and lipase ($P<0.05$) in the rumen of STH lambs. This may imply that appropriate doses of EOZB help ruminants to utilize plant-based feeds and enhance lipid metabolism.

The main function of members of the phylum Bacteroidetes in the host is to degrade carbohydrates and proteins [31, 32]. The main function of members of the phylum Firmicutes in the host is to enhance energy metabolism [33]. A high ratio of F/B (Firmicutes/Bacteroidetes) can help the host to efficiently absorb energy substances and maintain metabolic balance [34, 35]. In this study, Bacteroidetes, Firmicutes were the dominant group at the level of Phylum. Bacteroidetes significantly increased ($P<0.05$) at the dose of EOZB (10 ml/kg), Firmicutes significantly increased ($P<0.05$) at the dose of EOZB (5 ml/kg) and F/B (Firmicutes/Bacteroidetes) significantly increased ($P<0.05$) at the dose of EOZB (5 ml/kg). ($P<0.05$). This may imply that appropriate doses of EOZB help ruminants digest and absorb carbohydrates and proteins, and enhance energy metabolism. In addition, it has been shown that Tenericutes is the most prevalent bacterial phylum in diseased crabs [36]. Spirochaetes is a highly prevalent pathogenic agent [37]. EOZB significantly reduced the relative abundance of Tenericutes and Spirochaetes ($P<0.05$). This may imply that EOZB has certain antimicrobial properties that help the animal organism to resist pathogenic bacteria.

*Prevotella* has properties that degrade plant polysaccharides and promote inflammatory processes [38, 39]. At the genus level, the dominant genera in the STH lambs rumen were *Prevotella_1* and *Rikenellaceae_RC9_gut_group*. We found that the relative abundance of *Prevotella_1* was significantly lower ($P<0.05$) in the trial group supplemented with EOZB, which may imply that EOZB has anti-inflammatory properties. The specific function of the *Rikenellaceae_RC9_gut_group* is currently unknown, but the *Rikenellaceae_RC9_gut_group* is closely related to members of the genus *Alistipes* [40], and *Alistipes* has the effect of degrading plant-based polysaccharides [41, 42]. Therefore, the *Rikenellaceae_RC9_gut_group* may be associated with the degradation of phytogenic polysaccharides. In addition, *Rikenellaceae* resists the development of colonic inflammation and can limit the inflammatory process by stimulating T-cell differentiation [43]. We found that the relative abundance of

*Rikenellaceae_RC9_gut_group* was significantly increased (*P*<0.05) at the dose of EOZB (10 ml/kg), which may suggest that EOZB has the function of helping ruminants to utilize plant-based carbohydrates and inhibit inflammatory processes. Cellulolytic bacteria are an important group of bacteria that degrade cellulose in the rumen and play a key role in the production of VFA [44, 45]. Among these Cellulolytic bacteria, *Fibrobacter* increased significantly (*P*<0.05) at the dose of EOZB (15 ml/kg) and the *Ruminococcaceae_NK4A214_group* increased significantly (*P*<0.05) at the dose of EOZB (10 ml/kg). This may imply that EOZB tools have a positive effect on the utilization of cellulose-based nutrients by ruminants.

We used PICRUSt2 to predict rumen microbial gene function and to reveal the influence of EOZB on rumen microbial function. The prediction of the KEGG gene family indicated that the addition of EOZB could significantly enhance functions related to the Circulatory system (P = 0.009). EOZB (5 ml/kg and 10 ml/kg) significantly enhance the functions related to Infectious diseases: Viral. (P = 0.019, P = 0.034). The prediction of the COG gene family indicated that adding EOZB could significantly enhance functions related to Lipid transport and metabolism (P = 0.038). The enhancement of these functions in STH lambs may be related to the medicinal properties and therapeutic dose of EOZB, but there are currently no relevant reports of such findings.

Studies have shown that metabolite changes, microbial changes, and their connections are the main risk factors for abnormal tissue function and diseases, such as cancer [46], metabolic diseases [47], cardiovascular diseases [48], insulin resistance [49], and obesity [50]. Changes in gastrointestinal microbiota can directly alter their metabolic capacity and then affect local gastrointestinal function. For example, the fecal microbiome and metabolome of patients with systemic lupus erythematosus are simultaneously disordered [51]. The results of the current study showed that EOZB changed rumen protein digestion and absorption and glycerophospholipid metabolism in STH lambs. The essential amino acid tryptophan (TRP), as a precursor of protein synthesis, serotonin synthesis, and the kynurenine (KYN) pathway, plays a vital role in different physiological processes in the body and is closely related to the immune system and nervous system. The vast majority of available tryptophan is metabolized through the KYN pathway [52]. This pathway can promote the release of inflammatory cytokines IFN-γ and IL-6. Furthermore, KYN can be converted into kynurenic acid (KA) or nicotinamide adenine dinucleotide+ (NAD+) under the action of KYN transferase [53]. The immunomodulatory effect of the KYN pathway is related to the conversion level of KYN and the expression level of KA. Both KYN and KA can promote the differentiation of T cell subsets into regulatory T cells [54]. KYN molecules can also be decomposed into nicotinamide adenine dinucleotide+ (NAD+) and quinolinic acid (QA) through the KYN pathway. In the central nervous system, QA mediates neuronal excitotoxicity, while kynurenic acid provides neuronal protection. However, only TRP and KYN can cross the blood-brain barrier, while QA and KA cannot [55]. Therefore, the kynurenine pathway has the effect of regulating the immune system and nervous system, and the tryptophan-kynurenine pathway may be used to explain the brain-gut axis theory of microbiota and hosts. Interestingly, in this study, the rumen chyme of STH sheep in the EOZB group (BLW) was analyzed by metabolomics, and the expression of the differential metabolite kynurenic acid was higher than that of the control group (ALW). This may suggest that EOZB can modulate the immune system by enhancing the kynurenine pathway. It has been widely shown that phosphatidylcholine is a powerful marker of age-related membrane degeneration and low levels in plasma are positively associated with cognitive decline [56–59]. The main components of phosphatidylcholine include the long-chain polyunsaturated fatty acid docosahexaenoic acid (DHA) and choline, which can predict age-related executive function decline [60–63]. Interestingly, pc(18:3(6z,9z,12z)/15:0) expression in this study was significantly lower in the CLW group than in the control group (*P*<0.05), which

may imply that low doses of EOZB can inhibit the nervous system and thus have an analgesic pharmacological function.

The host gastrointestinal microbiota can help the host digest and absorb various nutrients, and at the same time produce various small molecules that affect the host's physiological health, immunity, metabolism, growth, development, etc. [64]. It has been shown that the microbiome and metabolome of colorectal cancer patients are simultaneously disturbed [65–67]. Thus, there is a link between the microbiota and the small molecules it produces. It has been shown that *Prevotella* has properties that promote the inflammatory process [68]. Similarly, 2_piperidinone has biological activities in organisms that promote angiogenesis and promote the development of inflammation [69]. We found that *Prevotella_1* was significantly and positively correlated with 2_piperidinone (r = 0.778, *P*<0.001). Moreover, the abundance of *Prevotella_1* and the expression of 2_piperidinone were both significantly lower in the trial group supplemented with EOZB than in the control group. This further confirms that EOZB has anti-inflammatory properties. It has been reported that the relative abundance of *Christensenellaceae* is negatively correlated with body weight [70]. The tryptophan kynurenine pathway promotes the release of inflammatory cytokines, which in turn promotes the inflammatory process [71]. And indole is one of the end products of the tryptophan kynurenine pathway [72]. We found that *Christensenellaceae_R_7_group* was significantly negatively correlated with indole (r = -0.492,*P*<0.001). Moreover, *Christensenellaceae_R_7_group* abundance was significantly increased (*P*<0.05) at EOZB (5 ml/kg) dose, while indole content was significantly decreased (*P*<0.05) at EOZB (5 ml/kg and 15 ml/kg) doses. This may imply that EOZB has anti-inflammatory properties. Many digestive enzymes in the rumen of ruminants are provided by microbiota, such as cellulases, amylases, etc. [73]. This suggests a link between the rumen microbiota and digestive enzymes. We found that *Christensenellaceae_R_7_group* was significantly positively correlated with lipase (r = 0.694, *P*<0.001). This may imply that EOZB can alter the activity of certain enzymes by changing the abundance of rumen microbiota, which in turn helps ruminants digest and absorb nutrients.

## Conclusions

In summary, the addition of specific doses of EOZB to the diet affected rumen enzyme activity, microbiome, metabolome, and the correlation between microbiota with metabolite and enzyme activities in STH lambs. The interactions between certain metabolites and enzyme activity with microbiota composition in the rumen of STH lambs could better explain their function, and the correlations identified could reveal which physiological processes EOZB regulates in STH lambs and provide new insights for similar studies in the future.

## Supporting information

**S1 Table. Sample sequencing data statistics.**
(DOCX)

**S2 Table. Bacterial abundance at the phylum and genus level.**
(XLSX)

**S3 Table. All differential metabolites.**
(XLSX)

**S4 Table. Differential metabolites between groups.**
(XLSX)

## Acknowledgments

We thank the staff at Sen Wuzhu Field and Pasture Farming Cooperative for helping to feed the test animals! We also thank the Gansu Agriculture University and the Gansu Academy of Agricultural Sciences for their help in collecting samples!

## Author Contributions

**Data curation:** Hailong Zhang, Xia Lang.

**Formal analysis:** Hailong Zhang, Xiao Li, Cailian Wang.

**Funding acquisition:** Xia Lang, Guoshun Chen, Cailian Wang.

**Investigation:** Hailong Zhang, Xia Lang, Xiao Li, Cailian Wang.

**Methodology:** Hailong Zhang, Guoshun Chen.

**Resources:** Hailong Zhang.

**Software:** Hailong Zhang.

**Writing – original draft:** Hailong Zhang.

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
