## [Decision Letter · Decision Letter 0]

29 Jun 2022

PONE-D-22-08057Effect of Zanthoxylum bungeanum essential oil on rumen enzyme activity, microbiome, and metabolites in lambsPLOS ONE

Dear Dr. Wang,

Thank you for submitting your manuscript to PLOS ONE. After careful consideration, we feel that it has merit but does not fully meet PLOS ONE’s publication criteria as it currently stands. Therefore, we invite you to submit a revised version of the manuscript that addresses the points raised during the review process.

We look forward to receiving your revised manuscript.

Kind regards,

Chun Wie Chong

Academic Editor

PLOS ONE

Journal Requirements:

https://journals.plos.org/plosone/s/file?id=ba62/PLOSOne_formatting_sample_title_authors_affiliations.pdfv

We thank for the financial support: No. 1760683; No.18YF1NA091; No. 18JR2RA032;

We thank for the financial support: No. 1760683; No.18YF1NA091; No. 18JR2RA032;

However, funding information should not appear in the Acknowledgments section or other areas of your manuscript. We will only publish funding information present in the Funding Statement section of the online submission form. 

We thank for the financial support: No. 1760683; No.18YF1NA091; No. 18JR2RA032;

The authors declare that the research was conducted in the absence of any commercial or financial relationships that could be construed as a potential conflict of interest.

Reviewers' comments:

Reviewer's Responses to Questions

**Comments to the Author**

1. Is the manuscript technically sound, and do the data support the conclusions?

Reviewer #1: Yes

Reviewer #2: Yes

2. Has the statistical analysis been performed appropriately and rigorously? 

Reviewer #1: Yes

Reviewer #2: Yes

3. Have the authors made all data underlying the findings in their manuscript fully available?

Reviewer #1: Yes

Reviewer #2: Yes

4. Is the manuscript presented in an intelligible fashion and written in standard English?

Reviewer #1: Yes

Reviewer #2: Yes

5. Review Comments to the Author

Reviewer #1: Figure 1D: ANOSIN (R na P) result are too small. Please provide a better figure.

Table 3. It would be best to standardize two digits after the dot

Line 277: How can you explain the predict genes of “Circulation system” , “Endrocne metabolic desease, Cytoesqueleton,....” if you provided the Microbial DNA sequencing?

LINE 285: The figure 4 legend needs to be improve.

Line 287: The same as before. Legend needs to better described.

Line 370: The same as before. Legend needs to better described

Reviewer #2: Line 20: The description "Subgroups:" sounds a bit strange, please check with the author.

Line 44: The numbers in the "CH4" area should be subscripts.

Line 72-74：What is the basis for the supplemental levels of EOZB of 0, 5, 10 and 15mg/kg in the experimental design? The trial lasted 52 days，why did choose this way?

Line 80-81: How was EOZB added? It is not stated in the dietary formula.

Line198-199：Which multiple-comparison is used?

Line213：Standard deviation rather than standard error should be used in Table 2.

Line 212,214：The P should be in italics.

Line 355, 342, 345, 351, 353, 354, 357, 359, 360: Spearman's analysis r value should be -0.5-0.5,please ask the authors to verify.

Line 377: References should be marked after "anti-inflammatory".

Line 388: References should be marked after the word "antioxidant".

6. PLOS authors have the option to publish the peer review history of their article (what does this mean?). If published, this will include your full peer review and any attached files.

Reviewer #1: No

Reviewer #2: No

---

## [Author Response · Author response to Decision Letter 0]

1 Jul 2022

Reviewer #1: Figure 1D: ANOSIN (R na P) result are too small. Please provide a better figure.

Re: We have replaced the figure and it is a little bigger in the "R and P" section.

Table 3. It would be best to standardize two digits after the dot

Re: We have modified the two digits after the dot and marked it in Table 3.

Line 277: How can you explain the predict genes of “Circulation system” , “Endrocne metabolic desease, Cytoesqueleton,....” if you provided the Microbial DNA sequencing?

Re: After the OTU abundance tables were normalised, the relationships corresponding to each OTU were compared to the KEGG and COG libraries, using PICRUSt2 to obtain the KO information and COG family information corresponding to the OTU. And calculate the abundance of each KO and the abundance of COG. Based on the information compared to the KEGG database, KO, Pathway and EC information can be obtained and the abundance of each functional class can be calculated based on the OTU abundance.

LINE 285: The figure 4 legend needs to be improve.

Re: We have modified and marked it in the article.

Line 287: The same as before. Legend needs to better described.

Re: We have modified and marked it in the article.

Line 370: The same as before. Legend needs to better described

Re: We have modified and marked it in the article.

Reviewer #2: Line 20: The description "Subgroups:" sounds a bit strange, please check with the author.

Re: We have modified and marked it in the article.

Line 44: The numbers in the "CH4" area should be subscripts.

Re: We have modified and marked it in the article.

Line 72-74：What is the basis for the supplemental levels of EOZB of 0, 5, 10 and 15mg/kg in the experimental design? The trial lasted 52 days，why did choose this way?

Re: Four gradients were established with " In vivo study on the efficacy of essential oil of Zanthoxylum bungeanum pericarp in dextran sulfate sodium-induced murine experimental colitis" and the cost of EOZB in order to select the amount of addition that would be beneficial to the healthy growth of lambs while controlling the cost. The trial period of 52 days is set based on the time limit of local lambs fattening, which is easy to combine with production practice.

Line 80-81: How was EOZB added? It is not stated in the dietary formula.

Re: A specific dose of EOZB was mixed with the concentrate to make pellets. We have modified and marked it in the article.

Line198-199：Which multiple-comparison is used?

Re: We used Duncan's method for multiple comparisons of components. We have modified and marked it in the article.

Line213：Standard deviation rather than standard error should be used in Table 2.

Re: We have modified in Table 2 and marked.

Line 212,214：The P should be in italics.

Re: We have modified all P in the full article to italics and marked it.

Line 355, 342, 345, 351, 353, 354, 357, 359, 360: Spearman's analysis r value should be -0.5-0.5, please ask the authors to verify.

Re: The range of Spearman's analysis r value is indeed -0.5-0.5. We have modified and marked it in the article.

Line 377: References should be marked after "anti-inflammatory".

Re: We have added references and revised the numbering accordingly. "[24]" is marked in the article.

Line 388: References should be marked after the word "antioxidant".

Re: We have added references and revised the numbering accordingly. "[25]" is marked in the article.

Other questions

Re: We have revised the format according to the requirements.

Re: We have revised this section, and provided the funder's role statement.

Role of Funder statement: The funder, Cailian Wang, played an important role in the study design design. The funder, Guoshun Chen, played an important role in the decision to publish.

3. However, funding information should not appear in the Acknowledgments section or other areas of your manuscript. We will only publish funding information present in the Funding Statement section of the online submission form. Please remove any funding-related text from the manuscript and let us know how you would like to update your Funding Statement.

Re: We have modified the ACKNOWLEDGMENTS section.

Funding Statement：We thank for the financial support: No. 1760683; No.18YF1NA091; No. 18JR2RA032;

4. Please complete your Competing Interests on the online submission form to state any Competing Interests. If you have no competing interests, please state "The authors have declared that no competing interests exist."

Re: We have modified the Conflicts of Interest section.

Author Statement: The authors have declared that no competing interests exist.

5. PLOS requires an ORCID iD for the corresponding author in Editorial Manager on papers submitted after December 6th, 2016. Please ensure that you have an ORCID iD and that it is validated in Editorial Manager.

Re: We have completed this operation.

6. Please include captions for your Supporting Information files at the end of your manuscript, and update any in-text citations to match accordingly.

Re: We have added the Supporting Information at the end of the article.

7. Please review your reference list to ensure that it is complete and correct.

Re: We checked our references again and it was compliant.

---

## [Editor Report · Decision Letter 1]

18 Jul 2022

Effect of Zanthoxylum bungeanum essential oil on rumen enzyme activity, microbiome, and metabolites in lambs

PONE-D-22-08057R1

Dear Dr. Wang,

We’re pleased to inform you that your manuscript has been judged scientifically suitable for publication and will be formally accepted for publication once it meets all outstanding technical requirements.

Kind regards,

Chun Wie Chong

Academic Editor

PLOS ONE
---

## [Editor Report · Acceptance letter]

20 Jul 2022

PONE-D-22-08057R1 

Effect of Zanthoxylum bungeanum essential oil on rumen enzyme activity, microbiome, and metabolites in lambs 

Dear Dr. Wang:

I'm pleased to inform you that your manuscript has been deemed suitable for publication in PLOS ONE. Congratulations! Your manuscript is now with our production department. 

Kind regards, 

on behalf of

Dr. Chun Wie Chong 

Academic Editor

PLOS ONE